# Decanoic Acid Stimulates Autophagy in *D. discoideum*

**DOI:** 10.3390/cells10112946

**Published:** 2021-10-29

**Authors:** Eleanor C. Warren, Pavol Kramár, Katie Lloyd-Jones, Robin S. B. Williams

**Affiliations:** Centre for Biomedical Sciences, Department of Biological Sciences, Royal Holloway University of London, Egham TW20 0EX, UK; Eleanor.Warren.2016@live.rhul.ac.uk (E.C.W.); Pavol.Kramar.2017@live.rhul.ac.uk (P.K.); Katie.Lloyd-Jones@rhul.ac.uk (K.L.-J.)

**Keywords:** autophagy, cancer, decanoic acid, *Dictyostelium*, epilepsy, 4BCCA, 4EOA, medium-chain triglycerides

## Abstract

Ketogenic diets, used in epilepsy treatment, are considered to work through reduced glucose and ketone generation to regulate a range of cellular process including autophagy induction. Recent studies into the medium-chain triglyceride (MCT) ketogenic diet have suggested that medium-chain fatty acids (MCFAs) provided in the diet, decanoic acid and octanoic acid, cause specific therapeutic effects independent of glucose reduction, although a role in autophagy has not been investigated. Both autophagy and MCFAs have been widely studied in *Dictyostelium*, with findings providing important advances in the study of autophagy-related pathologies such as neurodegenerative diseases. Here, we utilize this model to analyze a role for MCFAs in regulating autophagy. We show that treatment with decanoic acid but not octanoic acid induces autophagosome formation and modulates autophagic flux in high glucose conditions. To investigate this effect, decanoic acid, but not octanoic acid, was found to induce the expression of autophagy-inducing proteins (Atg1 and Atg8), providing a mechanism for this effect. Finally, we demonstrate a range of related fatty acid derivatives with seizure control activity, 4BCCA, 4EOA, and Epilim (valproic acid), also function to induce autophagosome formation in this model. Thus, our data suggest that decanoic acid and related compounds may provide a less-restrictive therapeutic approach to activate autophagy.

## 1. Introduction

Autophagy is a key homeostatic process required to degrade cytosolic proteins and organelles, with alterations in this pathway associated with a range of neurological disorders. Evidence suggests that impaired autophagy contributes to epileptogenesis [1,2,3,4], and has been implicated in both Alzheimer’s [5,6] and Parkinson’s disease [7,8,9]. Thus, new therapeutic approaches for the treatment of these diseases may involve mechanisms to upregulate autophagy.

Dietary therapies, such as ketogenic diets, are widely considered to be safe and inexpensive methods of activating autophagy. This process has been shown in the treatment of epilepsy [10,11], where the classical ketogenic diet, involving a heavily reduced dietary intake of carbohydrates in combination with a high consumption of long-chain fatty acids, is thought to activate autophagy in a similar manner to starvation [10,12,13,14]. In addition, ketone bodies generated by this diet have been suggested to stimulate neuronal autophagy and mediate neuroprotection [15]. More recently, there is increasing interest in an alternative less-restrictive form of the ketogenic diet called the medium-chain triglyceride (MCT) diet. This diet provides energy in the form of medium-chain fatty acids, mainly octanoic acid (60%) and decanoic acid (40%), which allow improved palatability and equivalent seizure control [16,17].

The therapeutic mechanism of the MCT diet has recently been proposed through activity of decanoic acid, initially from studies based in *Dictyostelium* [18], but then validated in mammalian models [19,20]. Multiple therapeutic mechanisms have been described for decanoic acid, including the inhibition of phosphoinositide signaling [18], the inhibition of diacylglycerol kinase [21], the direct inhibition of α-amino-3-hydroxy-5-methyl-4-isoxazolepropionic acid receptor (AMPA) receptors [20,22,23], and the activation of peroxisome proliferator-activated receptor gamma (PPAR-γ) [24]. Although these studies highlight molecular roles for decanoic acid rather than octanoic acid, oxidation of both fatty acids in neurons favors octanoic acid, leading to elevated levels of decanoic acid when provided in combination [25]. However, the ability of these fatty acids to modulate autophagy has not yet been examined. Here, we employ *Dictyostelium* as a well-established model organism in the field of epilepsy research [18,26,27,28,29,30] to investigate the effects of medium-chain fatty acids on autophagy.

Macroautophagy, the most prevalent form of autophagy, is conserved between *Dictyostelium* and humans. In *Dictyostelium,* macroautophagy (hereafter referred to as autophagy) is required to liberate nutrients during starvation, as well as in the degradation of proteins, and in response to pathogen infection [31,32,33]. *Dictyostelium* has been widely used as a model to study autophagy with findings from this organism providing important advances in the study of autophagy-related pathologies [34,35,36,37]. Research in *Dictyostelium* has demonstrated a link between the neurodegenerative disease, Chorea-acanthocytosis (ChAc), and autophagy [38], with further research implicating Vmp1, an ER protein involved in cancer, in the clearance of ubiquitinated protein aggregates through autophagy [39]. In addition to this, several studies in *Dictyostelium* have suggested a role for protein sequestration in neuronal malfunction [40,41]. The prevalence of autophagy-related studies using *Dictyostelium* highlights the value of this model to research this conserved cellular process.

Due to the widespread use of *Dictyostelium* in autophagy research, there are several well-established methods for assessing autophagy in this model. Fluorescence microscopy is widely employed to visualize autophagy structures in the cell, with Atg8 (microtubule-associated protein 1A/1B-light chain 3 (LC3) in mammals) tagged to GFP (GFP-Atg8) used extensively in *Dictyostelium* as a fluorescent marker [33]. GFP-Atg8 is employed as an autophagy marker since this protein is incorporated into the membrane of the phagophore following the induction of autophagy and remains in the membrane until the autophagosome is degraded [42,43]. An increase in the number of GFP-Atg8 positive structures in the cell indicates an induction in autophagy, whereas a decrease in the number and an increase in the size of GFP-Atg8-labeled structures suggests a blockage of autophagy [44,45]. Another method to study autophagy involves monitoring the amount of material degraded by this pathway over time, termed autophagic flux [33]. Autophagic flux is commonly measured by observing the autophagic cleavage of GFP from tagged cytosolic proteins such as GFP-Atg8 and the subsequent degradation of GFP [46]. Quantification of these proteins by Western blot provides an effective indication of changes in autophagic flux [32,47]. One further method to study autophagy is to measure expression levels of autophagy-related genes, using quantitative reverse transcription polymerase chain reaction (qRT-PCR) [32]. The availability of multiple well-studied techniques, along with the conservation of most mammalian autophagy genes, makes *Dictyostelium* an ideal model for the study of autophagy.

In this study, we demonstrate that decanoic acid activates autophagy in *Dictyostelium*, initially through a significant increase in the number of autophagosomes, visualized using live cell fluorescent microscopy to quantify GFP-Atg8 positive structures. We further demonstrate that decanoic acid increases autophagic flux and that decanoic acid treatment leads to an increase in the expression of two autophagy genes in wild type cells, suggesting a role for medium-chain fatty acids in inducing autophagy in *Dictyostelium*. We further assess the effects of fatty acid derivatives on autophagy in *Dictyostelium* to establish that a range of related fatty acid derivatives with established seizure control activity also function to induce autophagy in this model.

## 2. Materials and Methods

### 2.1. Autophagosome Formation Analysis

Cells expressing GFP-Atg8 were set up at a density of 1.33 × 10^6^ cells/mL in 2 mL of HL5 media (Formedium, Hunstanton, Norfolk, UK, HLB0103) in six-well plates. Cells were treated with decanoic acid (60 μM), octanoic acid (60 μM), or DMSO control for 22 or 2 h to provide final treatment durations of 24 and 4 h. Autophagy inducer AR-12 (Selleckchem, Houston, TX, USA, OSU-03012), or DMSO control, was added, and shaking was continued for 1 h before addition of protease inhibitors (Roche, Basel, Switzerland, 11873580001) at a final concentration of 2.5-fold or HL5 control. Shaking was continued for 1 h before cells were removed (without additional pipetting) to tubes on ice. For live-cell imaging, cells were centrifuged for 3 min at 500 g at 4 °C and resuspended in KK2 (16.2 mM KH_2_PO_4_ and 4 mM K_2_HPO_4_, pH 6.1) and were imaged under a layer of 1% KK2 agar. Time in KK2 and time under agar were kept constant between each condition (20 min in KK2 and 3 min under agar). Cells were imaged on an Olympus IX71 wide-field fluorescence microscope. Images were captured using a Micropix camera, model Elite 2. For measurement of Atg8-positive structure size, cells were analyzed using ImageJ, and cell size was measured across the largest diameter. Thirty cells were analyzed per experiment.

### 2.2. Autophagic Flux Analysis

Cells expressing GFP-Atg8 were treated as described above. Following removal from the six-well plate, cells were centrifuged for 3 min at 500× *g* at 4 °C and lysed in RIPA buffer (1 × 10^8^ cells/mL) containing protease inhibitor (Roche, Basel, Switzerland, 04693159001), before mixing 1:1 in 2× laemmli buffer resulting in 5 × 10^7^ cells/mL. Lysates were boiled in laemmli at 95 °C for 5 min before vortexing and centrifuging at max speed for 5 min. Samples (8 µL) were fractioned using a 12.5% gel and transferred to a PVDF membrane (Millipore, Watford, Hertfordshire, UK, IPFL00010) by wet transfer using transfer apparatus according to the manufacturer’s protocols (Bio-Rad, Watford, Hertfordshire, UK). After incubation with 5% BSA (Thermo Fisher Scientific, Loughborough, UK, AM2616) in PBS (Thermo Fisher Scientific, Loughborough, UK, 10010023) for 60 min, the membrane was incubated in 5% BSA in PBS with antibodies against GFP (Chromotek, Redcar, UK, 3H9, 1:1000) at 4 °C overnight. Membranes were washed in PBST and incubated with a 1:10,000 dilution of odyssey goat anti-rat IR DYE 800 (Li-Cor Biosciences, Cambridge, UK) and 1 in 10,000 streptavidin (Thermo Fisher Scientific, Loughborough, UK) in 5% BSA in PBS for 1 h. Blots were washed in PBST and visualized using the Odyssey CLx imager (Li-Cor Biosciences, Cambridge, UK).

### 2.3. PIP_3_ Production Assay

Cells expressing PHcrac-GFP were made chemotactically competent as described [48]. Briefly, cells were washed twice in development buffer (DB) before being resuspended in DB to 2 × 10^7^ cells/mL and rotated for 1 h at 110 rpm. Cells were pulsed with 60 nM of cAMP (Sigma, Gillingham, Dorset, UK, A6885) every 6 min for 4 h, in the presence or absence of the compound (60 µM decanoic acid or octanoic acid). Cells were basalated by the addition of caffeine (Sigma, Gillingham, Dorset, UK, C0750) to 5 mM, with rotation at 200 rpm for 30 min, before being washed twice in cold DB buffer and resuspended to 2 × 10^7^ cells/mL. For time-lapse imaging of PIP_3_ production, 360 µL chemotactically competent cells expressing PHcrac-GFP at 5 × 10^5^ cells/mL were placed into a well of an eight-well chambered cover glass (PI3K inhibitor LY294002 (Selleckchem, Houston, TX, USA, S1105) was added to one sample at a final concentration of 100 µM) and left to adhere for 10 min. Cells were stimulated by adding cAMP, to a final concentration of 1 µM, and images were captured every 2 s for 1 min.

### 2.4. PKB Activation Assay

Chemotactically competent cells were prepared in the presence or absence of the compound (60 µM decanoic acid or octanoic acid) as described above, with the addition of PI3K inhibitor LY294002 (Selleckchem, Houston, TX, USA, S1105) to one sample at a final concentration of 100 µM for the final 15 min of the preparation. For cAMP stimulation, 0.5 mL of chemotactically competent cells at 2 × 10^7^ cells/mL were transferred to a beaker shaking at 150 rpm, and cAMP was added to a final concentration of 1 µM. At time points of 10, 20, 30, 60, and 120 s, 40 µL of cells were transferred into microfuge tubes containing 10 µL of 5 × SDS sample buffer and were heated to 95 °C for 5 min. The p-PKC (pan) antibody was used to detect the phosphorylation state of the activation loops of PKBR1 (T309) and PKBA (T278) [48]. Briefly, samples (5 µL) were fractioned using a 10% gel and transferred to a 0.45 μm polyvinylidene difluoride membrane (Millipore, Watford, Hertfordshire, UK, IPFL00010) by wet transfer at 90 Volts for 1 h. Membranes were blocked with an odyssey blocking buffer (Li-Cor Biosciences, Cambridge, UK, 927-50100) for 1 h. Following blocking the membranes were incubated in an odyssey blocking buffer (with 1:1000 Tween 20) with primary antibody against p-PKC (pan) (1:2000 Phospho-PKC (pan) (zeta Thr410) rabbit antibody, Cell Signaling Technology, Danvers, MA, USA, 2060) at 4 °C overnight. Streptavidin (1:5000 Streptavidin, Alexa Fluor 680 conjugate, Thermo Fisher, Loughborough, UK, S21378) was used as a loading control. Membranes were washed in TBST and incubated with a 1:10,000 dilution of IRDye 800CW goat anti-rabbit IgG secondary antibody (Li-Cor Biosciences, Cambridge, UK, 925-3221) in an odyssey blocking buffer (with 1:1000 Tween20) for 1 h. Blots were washed in TBST and visualized using the Odyssey CLx imager (Li-Cor Biosciences, Cambridge, UK). Blots were quantified using the LI-COR Image Studio. Due to the large quantity of time points, conditions had to be analyzed on separate gels; to ensure consistency between treatment conditions, data were normalized to the loading control (MCCC1).

### 2.5. RT-qPCR Analysis of Autophagy Genes

Exponentially growing cells were treated for 1, 4, or 24 h with either decanoic acid (60 μM), octanoic acid (120 µM), or solvent (DMSO) control. RNA was extracted from these samples using the RNeasy mini kit (Qiagen, Manchester, UK, 74104) according to the manufacturer’s protocol. DNA was removed from the RNA using the DNA-free kit (Invitrogen, Loughborough, UK, AM1906) according to the manufacturer’s protocol. cDNA was synthesized using the RevertAid First Strand cDNA Synthesis Kit using oligo (dT)18 primers (Thermo Fisher Scientific, Loughborough, UK, K1621) according to the manufacturer’s protocol. qPCR was carried out using primers within the genes for *atg8a* and *atg1*, and the gapdh gene was used as an internal control. qPCR was carried out using SYBR^®^ Green JumpStart Taq ReadyMix (Sigma, Gillingham, Dorset, UK, S4438), and relative quantification was carried out using the 2^−ΔΔCt^ method.

### 2.6. Statistical Analysis

Two-tailed non-parametric Mann–Whitney *t*-tests were used to assess the significance between independent samples from two groups. The Kruskal–Wallis test with Dunn’s post hoc test was used to test significance between three or more independent groups of data. Statistical analysis was carried out using GraphPad Prism Software.

## 3. Results

### 3.1. Decanoic Acid Induces Autophagosome Formation

We initially investigated a role for decanoic acid and octanoic acid in the induction of an autophagic response in *Dictyostelium*. In these experiments, the incorporation of the autophagosome-associated Atg8 protein, linked to a fluorescent marker, was monitored, where GFP-Atg8-labeled autophagosome can be rapidly assessed (Figure 1A). To validate this approach, we initially quantified the number of GFP-Atg8-positive structures under control (growth) conditions and following starvation (1 h), where autophagy is induced. The number of GFP-Atg8-positive structures increased from an average of 2.6 autophagosomes per cell to an average of 7.3 autophagosomes per cell following starvation (*p* ≤ 0.01) (Figure 1B,C). This increase in autophagosome number was independent of changes in autophagosome size (Figure 1B,D). Starvation-induced autophagosome number increase is therefore consistent with an induction of autophagy [45].

We then investigated the effects of decanoic acid and octanoic acid on inducing autophagy in *Dictyostelium* during growth, in high glucose conditions. To do this, we quantified the average number of GFP-Atg8 structures following acute (4 h) or chronic (24 h) treatment with these fatty acids or with an autophagy inducer or a protease inhibitor. In these experiments, we chose a concentration of decanoic acid (60 µM) that reduces mTORC1 activity [49] during growth to show a significantly increased number of GFP-Atg8 positive structures, from an average of 2.78 autophagosomes per cell to 5.49 autophagosomes per cell after 4 h (*p* < 0.001) and 5.96 autophagosomes per cell following 24 h (*p* < 0.001) (Figure 1E,F). Treatment with octanoic acid (also at 60 µM) did not increase the number of GFP-Atg8 positive structures at either time point (Figure 1E,F). Treatment with the autophagy inducer (AR-12, 2.5 µM) increased the average number of GFP-Atg8-containing structures to 6.32 autophagosomes per cell (*p* < 0.001) (Figure 1E,F). Protease inhibitors also elevated the number of GFP-Atg8-containing structures (Figure 1E,F). None of the treatment conditions changed the size of GFP-Atg8-containing structures (Figure 1E,G). Our data suggest that the increase in autophagosomes observed following treatment with decanoic acid is consistent with a structurally specific induction of autophagy in the absence of starvation.

### 3.2. Decanoic Acids Enhances Autophagic Flux

Since our data suggest that decanoic acid treatment induces the formation of autophagosomes, consistent with the activation of autophagy, we continued by analyzing the effect of decanoic and octanoic acid on autophagic flux. This assay involves observing the autophagy-dependent cleavage of GFP-Atg8 present in autophagosomes, to form GFP, and the subsequent degradation of GFP. Autophagic flux can be activated by the addition of autophagy inducers to enhance both the cleavage of GFP-Atg8 and the breakdown of free GFP, or blocked by the presence of protease inhibitors [32].

The effects of decanoic acid and octanoic acid on autophagic flux was analyzed in *Dictyostelium* through the measurement of free GFP and GFP-Atg8 levels following acute (4 h) or chronic (24 h) treatment. In these experiments, cells were treated with decanoic acid or octanoic acid (60 µM), or with an autophagy inducer (AR-12, 2.5 µM) or a protease inhibitor [32] (Figure 2). In this analysis, acute (4 h) and chronic decanoic acid (24 h) reduced both free GFP and GFP-Atg8 levels (Figure 2B,C), with a significant reduction in the ratio of free GFP: GFP-Atg8 (Figure 2D), consistent with the activation of autophagic flux. In comparison, acute and chronic octanoic treatment reduced free GFP levels (Figure 2B); however, the ratio of free GFP: GFP-Atg8 levels was only reduced in acute treatment (Figure 2D), suggesting a short-term transient change in autophagic flux caused by octanoic acid. Treatment with the autophagy inducer reduced free GFP, leading to a significant reduction in the ratio of free GFP: GFP-Atg8 (Figure 2D), indicating increased autophagic flux. Treatment with the protease inhibitor reduced free GFP levels without altering GFP-Atg8 levels consistent with a block in the initial cleavage of GFP-Atg8 leading to a reduction in the generation of free GFP [32] (Figure 2B–D). These results demonstrate that decanoic acid treatment provides a consistent increase in autophagic flux in *Dictyostelium*.

### 3.3. Decanoic Acid Does Not Reduce PIP_3_ Levels

To investigate a mechanism for the decanoic acid-dependent activation of autophagy, we initially focused on a key phosphoinositide, PIP_3_ (PtdInsP_3_), that plays a key role in inhibiting autophagy through the activation of mTORC1, with a well-established role in regulating autophagy [50,51]. This pathway provided a logical choice for mechanism; since decanoic acid has been suggested to attenuate phosphoinositide signaling [18], we investigated the effects of both fatty acids on PIP_3_ levels in *Dictyostelium*. These experiments involved developing cells to an early aggregation stage and using a single pulse of cAMP (1 µM) to induce PI3K activity to produce PIP_3_ on the cell membrane, and measured this production using membrane localization of a fluorescent marker protein containing a pH domain linked to GFP [42,52]. In wild type cells, inducing PI3K activation provided a transient localization of the marker from the cytosol to the membrane (Figure 3A), with a peak membrane localization (equivalent to peak PIP_3_ production) at around 10 s post induction. Pretreatment of cells for 60 min with either decanoic or octanoic acid (60 µM), a concentration that induced autophagy in these cells, did not significantly reduce PIP_3_ production visualized by PHcrac-GFP membrane localization (Figure 3A). Quantification of this effect indicated no significant effect on the maximum (peak) PIP_3_ production (Figure 3B) nor the total production of PIP_3_ (area under the curve) (Figure 3C). Strong inhibition was demonstrated using a PI3K inhibitor, LY290064 (100 µM), confirming the specificity of this assay. These data suggest that both decanoic acid and octanoic acid are unlikely to regulate autophagy through inhibiting PIP_3_ production.

### 3.4. Decanoic Acid Provides Limited Regulation of PKB Activity

The regulation of autophagy has also been demonstrated through the evolutionarily conserved protein kinase B (PKB) [53]. This protein is activated following PIP_3_ production but also by a range of other signaling pathways independent of PI3K [54], suggesting that medium-chain fatty acids may function through PKB regulation independent of PI3K activity. Thus, to analyze the effect of decanoic acid and octanoic acid on PKB activation, we monitored the phosphorylation (activation) of PKBR1 (T309), which is dependent upon mTORC2 activity [48,55,56], and the activation of PKBA (T278), which is dependent upon both TORC2 and PIP_3_ [48]. Thus, western blotting using antibodies against the activation loop phosphorylation sites can be used to rapidly assess PKB activation from both TORC2 and PI3K-signalling pathways. Using cells in early development, consistent with the experiments for the analysis of PIP_3_, and inducing activation of PKBA and PKBR1 by a single pulse of cAMP (1 µM), cell samples were taken at intervals and analyzed by western blot (Figure 4). In the absence of MCFAs, peak activation occurred 10–60 s after stimulation (Figure 4A) for both PKB enzymes, and both decanoic and octanoic acid did not grossly modify activation, although inhibition of PI3K blocked PKBA activation. Quantification of the effect of decanoic acid on PKBA activation identified reduced activation at a single time point (20 s) compared to the control, with no effect of octanoic acid, with inhibition of PI3K activity providing significantly reduced induction (from 10–30 s) and total activation (area under curve) (Figure 4B). Quantification of PKBR1 activation also showed no effect of either decanoic acid or octanoic acid, although inhibition of PI3K activity provided a small but significant reduction of induction (from 10–20 s), with no change in total activation with any treatment (area under curve) (Figure 4C). Thus, these data suggest that decanoic acid is unlikely to regulate autophagy through the inhibition of PKB activity.

### 3.5. Decanoic Acid Increases Expression of Autophagy-Inducing Proteins

To understand the mechanisms leading to the decanoic acid dependent induction of autophagosome formation and increase in autophagic flux, we investigated a transcription-based effect of decanoic acid on the expression levels of two genes encoding autophagy-related proteins. We chose to analyze *atg1* [57,58] and *atg8a* [43,59], where both encoded proteins have a role in inducing autophagy. In these experiments, wild type cells were treated with decanoic acid (60 µM) and a higher concentration of octanoic acid to maximize potential changes in expression (120 µM), for 1, 4, or 24 h, and gene expression was assessed using quantitative reverse transcription PCR analysis (Figure 5). The expression of both *atg1* and *atg8a* was not altered following 1 h decanoic acid treatment, was noticeable but not non-significantly increased after 4 h treatment, and was significantly upregulated following 24 h treatment (*p* ≤ 0.01 for both), consistent with a chronic induction of autophagy (Figure 5A,B) [60]. Octanoic acid treatment did not induce either *atg1* or *atg8a* expression at any time point. Thus, we have demonstrated a structural specific effect of decanoic acid to increases the expression of autophagy-associated genes, consistent with an induction of autophagy.

### 3.6. Structural Specificity of Fatty Acids on Dictyostelium Autophagy

Structural specificity of medium-chain fatty acids in seizure control has previously been demonstrated in *Dictyostelium* and then in mammalian models. This specificity was initially identified through the reproduction of a molecular mechanism shown for the current epilepsy treatment, Epilim (valproic acid) [18], with these compounds then demonstrated to potently block seizure activity [19,20,61]. We therefore investigate the effects of a range of other medium-chain fatty acids on autophagy induction in *Dictyostelium* (Figure 6A). Initially, analyzing a seven carbon backbone medium-chain fatty acid, 2MHA (Figure 6B,C), that does not show seizure control activity [19], did not provide an autophagosome induction effect similar to that of decanoic acid. Nonanoic acid, a nine-carbon-backbone straight-chain fatty acid, intermediate between decanoic acid and octanoic acid, did not show autophagy induction. In contrast, two other related compounds, 4BCCA and 4EOA, both showing strong seizure control activity [19,20,61], significantly increased autophagosome formation, as did the valproic acid (Figure 6B,C). None of the compounds altered autophagosome size (Figure 6D). These data suggest that structurally specific MCFA that show seizure control activity may function to induce autophagy.

## 4. Discussion

Autophagy, required for the recycling of misfolded proteins and damaged organelles, is a key cellular process involved in a wide range of neurodegenerative diseases, where activation of the process has been suggested to provide therapeutic effects [62,63]. Here, we demonstrate that decanoic acid, provided by the MCT diet [22], stimulates autophagy in the biomedical model *Dictyostelium*. In this model [64], decanoic acid leads to an increase in the number of autophagosomes per cell and an increase in autophagic flux, likely caused by an increase in the expression of two autophagy-linked genes, *atg1* and *atg8a* [43,57,58,59]. In contrast, octanoic acid provides no effect on autophagosome number or chronic autophagic flux and does not induce expression of autophagy genes, indicating the specificity of this effect. We thus identify that decanoic acid may induce autophagy and autophagic flux through a structurally specific effect on transcription.

Autophagy has a recognized role in epilepsy that was first established when rapamycin, an autophagy inducer, was shown to suppress seizures [1]. Subsequently, a blockage in autophagy has been identified in patients with genetic conditions giving rise to seizures, such as tuberous sclerosis complex (TSC) [65,66] and Lafora disease [4]. Thus, novel approaches are currently being examined to activate autophagy as a treatment for epilepsy [2,67,68]. Furthermore, the capacity of the classical ketogenic diet and intermittent fasting to increase starvation-induced autophagy has substantiated these dietary interventions as potential treatments for epilepsy [69]. Dysfunctional autophagy is also associated with the neurodegenerative disorders, Huntington’s, Parkinson’s, and Alzheimer’s disease, and activating autophagy has proven successful in models of these disorders [66,70,71]. Interestingly, a new MCT ketogenic diet containing enhanced levels of decanoic acid has recently been demonstrated to provide strong seizure control effects on a range of patients with drug-resistant epilepsy [72], in the absence of ketosis, suggesting a key role for decanoic acid-dependent effects. In addition, another new MCT ketogenic diet also been shown to provide a mild increase in cognitive function in patients with early-stage Alzheimer’ disease [73,74]. Our findings suggest that further analysis of a role for decanoic acid in activating autophagy may identify therapeutic benefits in epilepsy and neurodegenerative disorders.

The role of fatty acids in regulating autophagy for therapeutic effect remains poorly understood. Our findings that show a decanoic acid-dependent activation of autophagy in *Dictyostelium* are consistent with that shown in mouse models, where dietary medium-chain fatty acids restore suppressed autophagy [75]. In regards to long-chain fatty acids, the majority of research in this field focuses on palmitic acid (C16:0) and oleic acid (C18:1), where both these fats either inhibit autophagy [76,77,78] or activate it [77,79,80,81]. Short-chain fatty acids have also been suggested to regulate autophagy [82,83] with the length of fatty acid as well as the saturation status thought to alter the effect on autophagy [78]. Thus, although a range of fatty acids have been shown to regulate autophagy, our findings now include decanoic acid in this list, with structural specificity in activating this process.

Decanoic acid has recently been demonstrated to modulate a range of cellular processes, where some of these may underlie the effect shown here for autophagy induction. Our initial focus was on decanoic acid-dependent regulation of phosphoinositide signaling, first identified in *Dictyostelium* [18], and particularly PIP_3_, since this effect was also demonstrated for Epilim using in vitro and in vivo mouse models [84]. In *Dictyostelium*, decanoic acid did not significantly reduce maximum or total PIP3 levels, nor reduce the total activation of PKBA (partially dependent upon PIP_3_) [48], although it provided a small but significant reduction in PKBA activation 20 s after stimulation. This suggests a small effect on PIP3 production but may not be enough to increase autophagy levels. However, decanoic acid has recently been shown to reduce the activation of mTORC1 in *Dictyostelium*, and additionally in rodent hippocampal slices and in iPSC-derived human astrocytes [49]. We propose here that the likely mechanism of decanoic acid in upregulating autophagy is due to inhibition of mTORC1, to activate transcription of autophagy inducing genes (*atg1* and *atg8*), to activate autophagy. This mechanism is consistent with the structurally specific effect of decanoic acid (not octanoic acid) in both inhibiting mTORC1 and inducing autophagy, independent of the glucose restriction and insulin signaling associated with the classical ketogenic diets [13,85,86,87]. Further analysis of genes known to be regulated by mTOR in *Dictyostelium* would provide additional evidence for the involvement of this pathway [88].

In this paper, we have identified that the medium-chain fatty acid, decanoic acid, activates autophagy in *Dictyostelium*. Since activation of autophagy has been suggested as a potential treatment for epilepsy and a range of neurological and neurodegenerative disorders, we suggest that further investigation into the mechanism behind fatty-acid-induced autophagy activation could enhance our understanding of the benefits of medium-chain fatty acids in health and disease.

## Figures and Tables

**Figure 1 cells-10-02946-f001:**
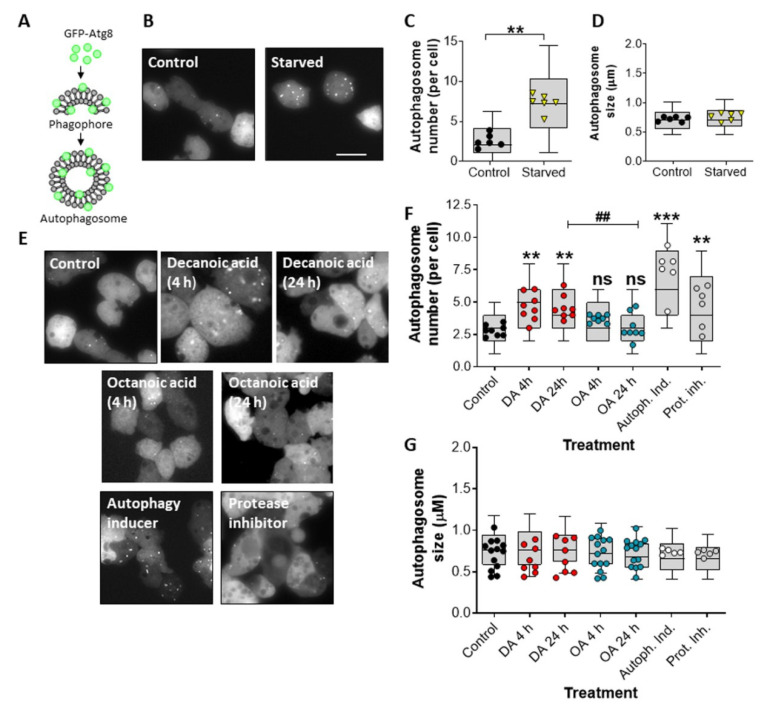
Decanoic acid induces autophagosome formation in *Dictyostelium*. (**A**) Exogenous GFP-Atg8 is integrated into the membranes of the phagophore and autophagosome to allow visualization of autophagy in cells. (**B**) Representative images of Dictyostelium cells expressing GFP-Atg8, untreated (control) or starved (1 h), and following analysis for (**C**) autophagosome number and (**D**) autophagosome size, shown as mean ± SEM (*n* = 6, Mann–Whitney *t*-test). (**E**) Representative images of *Dictyostelium* cells expressing GFP-Atg8 untreated (control, DMSO) or treated with decanoic acid (DA, 60 µM) or octanoic acid (OA, 60 µM) for 4 or 24 h, autophagy inducer (AR-12, 2.5 µM), or protease inhibitor (2.5×). Images were analyzed for (**F**) autophagosome number and (**G**) autophagosome size (*n* ≥ 6), shown as mean ± SEM (Kruskal–Wallis with Dunn’s multiple comparisons test). Significance against control is indicated by ns *p* > 0.05, ** *p* ≤ 0.01, *** *p* ≤ 0.001, or between decanoic acid and octanoic acid, ## *p* ≤ 0.01. Scale bars represent 20 µm. Each data point is derived from at least 30 individual cells or autophagosomes, with box and whisker plots showing 10–90th percentile range.

**Figure 2 cells-10-02946-f002:**
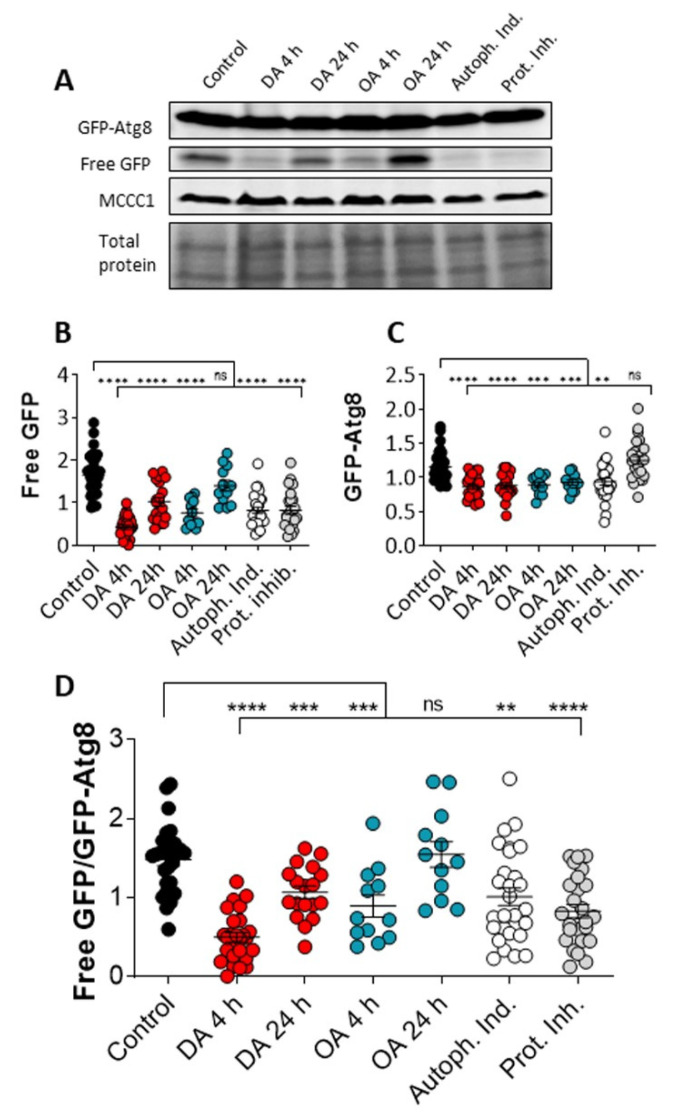
Decanoic acid enhances autophagic flux in *Dictyostelium*. Autophagic flux can be measured in *Dictyostelium* cells expressing GFP-Atg8 by western blot using a GFP-specific antibody to visualise total GFP-Atg8, cleavage to free GFP, and proteolytic degradation of free GFP, with MCCC1 as a loading control, and additionally visualized with total protein. Autophagic flux was analyzed in *Dictyostelium* cells, under control conditions (solvent-treated growing cells), or cells treated with decanoic acid (DA) or octanoic acid (OA), both at 60 µM, for 4 or 24 h, or with an autophagy inducer or protease inhibitor, showing (**A**) a representative western blot of GFP-Atg8, Free GFP, and loading control (MCCC1), and total protein levels. Quantification of (**B**) GFP-Atg8 and (**C**) free GFP levels, enabled the assessment of autophagic flux provided by (**D**) the ratio of free GFP:GFP-Atg8. Data, analyzed by a Kruskal–Wallis with Dunn’s multiple comparisons test, from *n* = 8, provides significance indicated by ns *p* > 0.05, ** *p* ≤ 0.01, *** *p* ≤ 0.001, and **** *p* ≤ 0.0001.

**Figure 3 cells-10-02946-f003:**
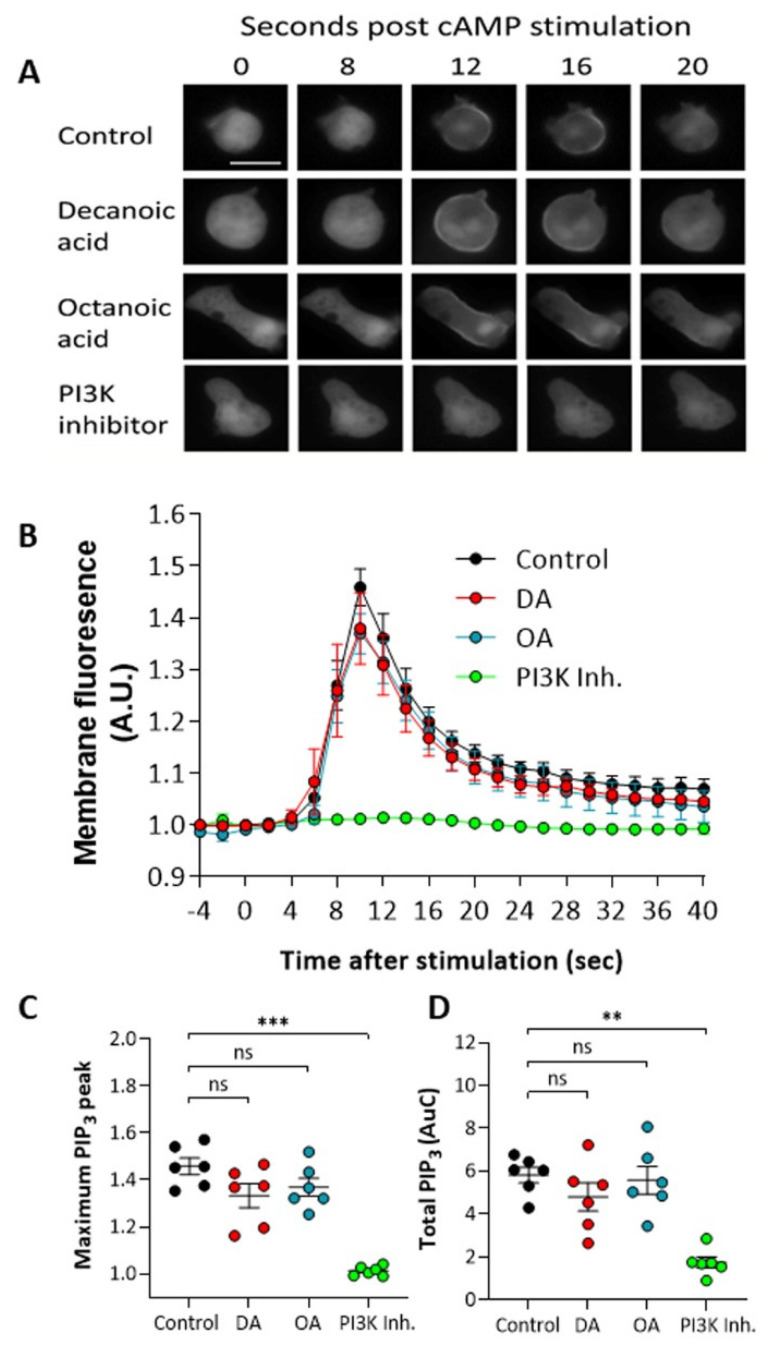
Medium-chain fatty acids do not reduce transient PIP_3_ production in *Dictyostelium* at concentrations that trigger autophagy. To visualize PIP_3_ production, cells expressing PHcrac-GFP in early development were treated with decanoic acid or octanoic acid (both at 60 µM, 4 h), PI3K inhibitor (100 µM LY294002, 15 min), or solvent control (DMSO) and stimulated with 1 µM cAMP to trigger production of PIP_3_ visualized by transient movement of PHcrac-GFP to the membrane. (**A**) Cells show transient movement of PHcrac-GFP to cell membranes, reflecting induction of PIP_3_ production, and (**B**) membrane fluorescence was quantified normalized to whole cell fluorescence, and (**C**) maximum membrane fluorescence (PIP_3_ production) and (**D**) total area under the curve (total PIP_3_ production) were compared between treatments. Data represent mean and SEM from six independent experiments (three individual cells were analyzed per experiment). Scale bar represents 10 µm. Significance is indicated by ns > 0.05, ** *p* ≤ 0.01, *** *p* ≤ 0.001 (Kruskal–Wallis with Dunn’s multiple comparisons test).

**Figure 4 cells-10-02946-f004:**
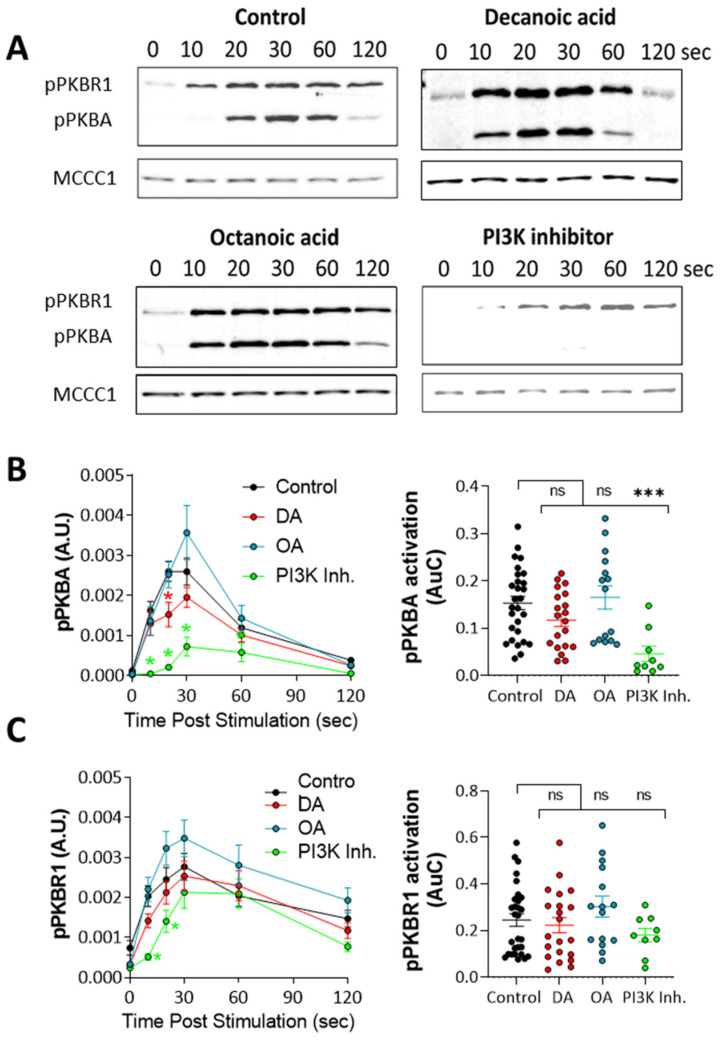
Medium-chain fatty acids do not reduce protein kinase B (PKB) phosphorylation in *Dictyostelium* at concentrations that trigger autophagy. To visualize the activation of PKB, cells in early development were treated with decanoic acid or octanoic acid (both at 60 µM, 4 h), PI3K inhibitor (100 µM LY294002, 15 min), or solvent control (DMSO) and stimulated with 1 µM cAMP to trigger transient phosphorylation of PKBA, and PKBR1 over 120 s was visualized by western analysis. (**A**) Under these conditions, cells indicated transient phosphorylation of both PKBA and PKBR1 compared to loading control (MCCC1). Quantification of (**B**) PKBA phosphorylation showed a small but significant decrease 20 s after induction in the presence of decanoic acid with no effect on total activation (area under the curve) and a large decrease in the presence of a PI3K inhibitor. Quantification of (**C**) PKBR1 phosphorylation showed a small but significant decrease in phosphorylation in the presence of a PI3K inhibitor, 20 s after induction, and a large decrease in total activation (area under curve). Data were derived from *n* ≥ 9 independent experiments. Significance is indicated by ns > 0.05, * *p* ≤ 0.05, *** *p* ≤ 0.001 (Kruskal–Wallis with Dunn’s multiple comparisons test).

**Figure 5 cells-10-02946-f005:**
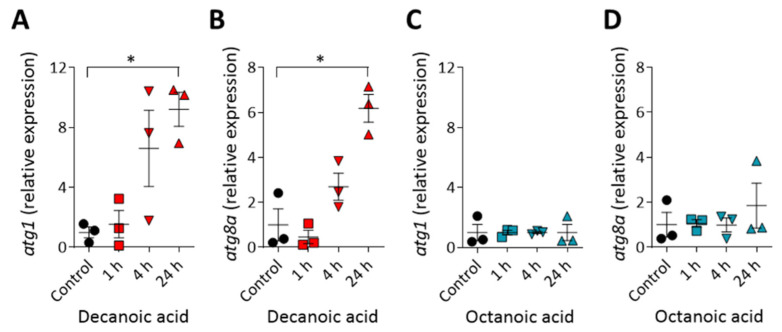
Decanoic acid but not octanoic acid increases autophagy gene expression after 24 h. Cells treated with decanoic acid (60 µM) (or control) were analyzed by RT-qPCR measuring the relative abundance of (**A**) *atg1* and (**B**) *atg8a*. Cells treated with octanoic acid (120 µM) (or control) were also analyzed by RT-qPCR measuring the relative abundance of (**C**) *atg1* and (**D**) *atg8a*. mRNA was normalized to the housekeeping gene *gapdh* in all conditions with the mRNA levels normalized so that untreated cells have a relative expression level of 1 (*n* = 3). Data represented are mean ± SEM (Kruskal–Wallis with Dunn’s multiple comparisons test). Significance is indicated by * *p* ≤ 0.05.

**Figure 6 cells-10-02946-f006:**
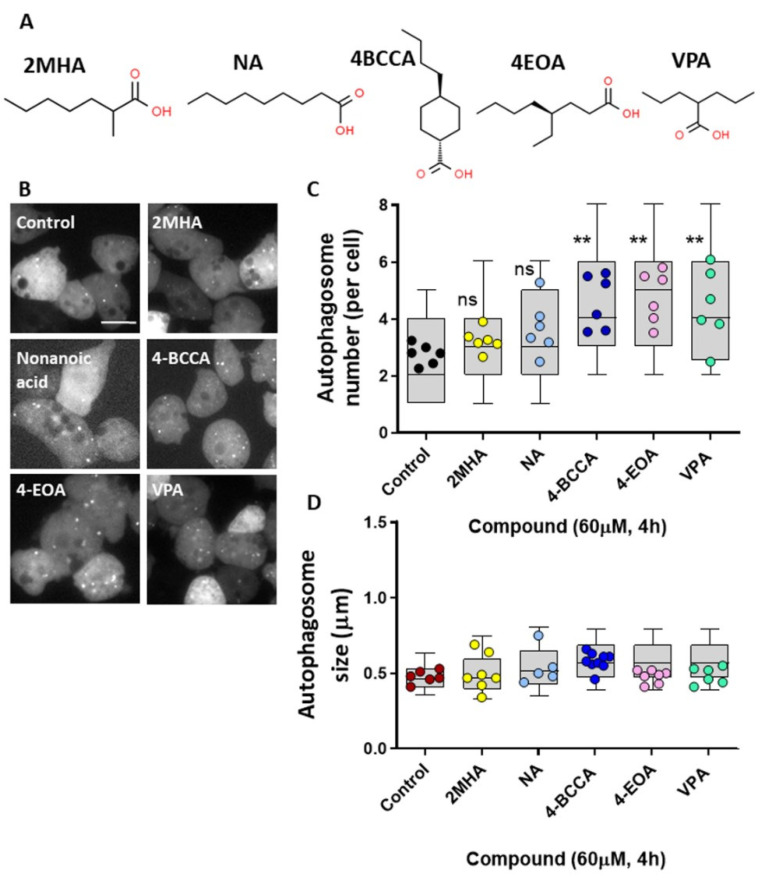
Specific MCFA shown to control seizures, and Epilim, induce autophagosome formation in *Dictyostelium*. (**A**) Chemical structures of the fatty acid, 2-methylheptanoic acid (2MHA), nonanoic acid (NA), 4-butylcyclohexane carboxylic acid (4BCCA), 4-ethyloctanoic acid (4EOA), and Epilim (VPA), where these compounds show variable activity in seizure control. (**B**) Representative images of *Dictyostelium* cells expressing GFP-Atg8 untreated (DMSO control) or treated with 60 µM of specified fatty acids or Epilim (4 h). Scale bar represents 10 µm. Images were analyzed for (**C**) autophagosome number per cell and (**D**) autophagosome size (*n* ≥ 5). Data represented are mean ± SEM (One-way ANOVA with Dunnets’s multiple comparisons test). Significance is indicated by ns *p* ≥ 0.05, ** *p* ≤ 0.01. Each data point is derived from 30 individual cells or autophagosomes, box and whisker (10–90th percentile).

## Data Availability

The datasets generate for this study are provided in the paper.

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
