# Peer review of "Decanoic Acid Stimulates Autophagy in D. discoideum"

_cells, 2021, doi:10.3390/cells10112946_

Round 1

Reviewer 1 Report

This paper describes that decanoic acid treatment induces autophagy in Dictyostelium cells through stimulation of transcription. The manuscript is well writing and well presented. The results are interesting and with the potential to be further extended.

I have three suggestions that may potentially enrich the MS content

1) Adjust the manuscript title since treatments were done by providing fatty acids directly and not through ketonic based diet

2) A detailed description of culture conditions and fatty acid treatment is missing.

3) A wide mTORC1 effect on transcription under several conditions is available, so including a couple of genes that are downregulated by mTOR will provide further evidence for the participation of this pathway in decanoic acid signaling. A good reference could be found in Jaiswal and Kimmel, BMC Biology 17,58 (2019)   

Author Response

We thank both reviewers for their positive response to the manuscript. We address the specific comments below. We attached the updated manuscript with 'track changes' to show the requested modifications.

Reviewer 1:

1) Adjust the manuscript title since treatments were done by providing fatty acids directly and not through ketonic based diet

We thanks the reviewer for this suggestion and have modified the title as requested.

2) A detailed description of culture conditions and fatty acid treatment is missing.

This has been corrected in the manuscript.

3) A wide mTORC1 effect on transcription under several conditions is available, so including a couple of genes that are downregulated by mTOR will provide further evidence for the participation of this pathway in decanoic acid signaling. A good reference could be found in Jaiswal and Kimmel, BMC Biology 17,58 (2019)   [1]

We thank the reviewer for this comment and have included this in the text.

Reviewer 2:

The authors investigated the effect of (mainly) decanoic acid and octanoic acid on Dictyostelium cells. The stimulation of autophagy by decanoic acid and increase of autophagic flux was demonstrated using a variety of established and validated experimental approaches. Several molecular pathways that could lead to this stimulation of autophagy were monitored (like PI3P), but showed either no or only a very limited (if at all) link to the stimulation induced by deconaoic acid. The stimulation of autophagy with decanoic acid was compared with the effect by other compounds and emphasized the structural specificty to decanoic acid.

The paper respresents a solid study employing established methods to investigate the effec ot medium chain fatty acids on autophagy and, in my view, only very minor corrections are neccessary.

The authors should make clear in the text that they are measuring the amount of mRNA (of atg8a and atg1) by qRT-PCR and are not simply performing quantitative PCR.

We have corrected this in the manuscript.

Please check the scale bars, especially for Figure 6.

Thank you for spotting this error. We have corrected it.

On page 11 the formatting was off which cut the last sentence of the paragraph (I assume it was the last sentence...).

We apologise for this – a formatting error in using the template for the journal. This has been corrected.

Please go through the text for uncomplete sentences etc.

We have carefully check the text.

Reviewer 2 Report

The authors investigated the effect of (mainly) decanoic acid and octanoic acid on Dictyostelium cells. The stimulation of autophagy by decanoic acid and increase of autophagic flux was demonstrated using a variety of established and validated experimental approaches. Several molecular pathways that could lead to this stimulation of autophagy were monitored (like PI3P), but showed either no or only a very limited (if at all) link to the stimulation induced by deconaoic acid. The stimulation of autophagy with decanoic acid was compared with the effect by other compounds and emphasized the structural specificty to decanoic acid.

The paper respresents a solid study employing established methods to investigate the effec ot medium chain fatty acids on autophagy and, in my view, only very minor corrections are neccessary.

The authors should make clear in the text that they are measuring the amount of mRNA (of atg8a and atg1) by qRT-PCR and are not simply performing quantitative PCR.

Please check the scale bars, especially for Figure 6.

On page 11 the formatting was off which cut the last sentence of the paragraph (I assume it was the last sentence...).

Please go through the text for uncomplete sentences etc.

Author Response

We thank the reviewer for their supportive comments. We address specific points below, and attach the modified paper showing 'track changes'. 

Reviewer 2:

The authors investigated the effect of (mainly) decanoic acid and octanoic acid on Dictyostelium cells. The stimulation of autophagy by decanoic acid and increase of autophagic flux was demonstrated using a variety of established and validated experimental approaches. Several molecular pathways that could lead to this stimulation of autophagy were monitored (like PI3P), but showed either no or only a very limited (if at all) link to the stimulation induced by deconaoic acid. The stimulation of autophagy with decanoic acid was compared with the effect by other compounds and emphasized the structural specificty to decanoic acid.

The paper respresents a solid study employing established methods to investigate the effec ot medium chain fatty acids on autophagy and, in my view, only very minor corrections are neccessary.

The authors should make clear in the text that they are measuring the amount of mRNA (of atg8a and atg1) by qRT-PCR and are not simply performing quantitative PCR.

We have corrected this in the manuscript.

Please check the scale bars, especially for Figure 6.

Thank you for spotting this error. We have corrected it.

On page 11 the formatting was off which cut the last sentence of the paragraph (I assume it was the last sentence...).

We apologise for this – a formatting error in using the template for the journal. This has been corrected.

Please go through the text for uncomplete sentences etc.

We have gone through the manuscript carefully and corrected errors.
